# Structural basis for negative regulation of the *Escherichia coli* maltose system

Yuang Wu [1,2,6], Yue Sun[3,6], Evelyne Richet[4], Zhifu Han[3] & Jijie Chai[1,2,3,5] ✉

Proteins from the signal transduction ATPases with numerous domains (STAND) family are known to play an important role in innate immunity. However, it remains less well understood how they function in transcriptional regulation. MalT is a bacterial STAND that controls the *Escherichia coli* maltose system. Inactive MalT is sequestered by different inhibitory proteins such as MalY. Here, we show that MalY interacts with one oligomerization interface of MalT to form a 2:2 complex. MalY represses MalT activity by blocking its oligomerization and strengthening ADP-mediated MalT autoinhibition. A loop region N-terminal to the nucleotide-binding domain (NBD) of MalT has a dual role in mediating MalT autoinhibition and activation. Structural comparison shows that ligand-binding induced oligomerization is required for stabilizing the C-terminal domains and conferring DNA-binding activity. Together, our study reveals the mechanism whereby a prokaryotic STAND is inhibited by a repressor protein and offers insights into signaling by STAND transcription activators.

To cope with a constantly changing environment, living organisms acquire different signaling machineries in response to both internal and external stimuli. The signal transduction ATPases with numerous domains (STAND) is a family of ubiquitous intracellular signaling proteins that play a key role in programmed cell death and innate immunity, exemplified by apoptotic proteins and NOD-like receptors (NLRs)[1–6]. In the presence of a death cue or pathogen invasion, these proteins can form multimeric complexes (animal apoptosomes and inflammasomes, plant resistosomes), which initiate downstream signaling cascades[7–12]. In bacteria, NLR-related proteins are involved in defense against phages[13,14]. A recent study shows that prokaryotic antiviral STAND (Avs) tetramerizes upon detection of hallmark viral proteins, leading to the activation of N-terminal effector domain and cell death, a process similar to RPP1 or Roq1-mediated cell death in plants[12,13,15]. Prokaryotic STANDs also comprise transcription regulators, which are important for monitoring the availability of nutrients and facilitating their efficient utilization[1].

MalT is the transcription activator of the maltose system of *Escherichia coli*, which orchestrates the uptake and metabolism of maltodextrins[16]. It bears all the hallmarks of STAND family proteins with a tripartite domain architecture, comprising an N-terminal nucleotide-binding and oligomerization (NOD) module, a super-helical peptide repeats (SUPR)-type sensor domain, and an effector domain that confers DNA-binding activity. While the transcriptional activity of MalT is induced by maltotriose, a sugar produced by internalization and metabolism of maltodextrins or degradation of intracellular glycogen[17], three repressor proteins that negatively modulate MalT activity have been identified[18–20]. MalK is the ATP-binding subunit of the maltose transporter. In the absence of substrate transport, MalK interacts with MalT and anchors it at the cytoplasmic membrane, thereby preventing its activation[21]. The other two MalT repressors are MalY a pyridoxal 5'-phosphate (PLP)-dependent dimeric enzyme and Aes, an enzyme with acetyl esterase activity. Both of them inhibit MalT through a direct interaction as shown in vitro, which could

[1]Institute of Biochemistry, University of Cologne, Cologne, Germany. [2]Max Planck Institute for Plant Breeding Research, Cologne, Germany. [3]Beijing Advanced Innovation Center for Structural Biology, Tsinghua-Peking Joint Center for Life Sciences, Center for Plant Biology, School of Life Sciences, Tsinghua University, Beijing, China. [4]Institut Pasteur, Université Paris Cité, CNRS UMR6047, INSERM U1306, Unité Biologie et génétique de la paroi bactérienne, Paris, France. [5]Key Laboratory of Structural Biology of Zhejiang Province, School of Life Sciences, Westlake University, Hangzhou, China. [6]These authors contributed equally: Yuang Wu, Yue Sun. ✉e-mail: chai@mpipz.mpg.de

be impeded by inducer binding[18,19,22,23]. Thus far, the physiological roles of MalT control by MalY and Aes are unclear.

MalT is known to cycle between two states, a monomeric ADP-bound resting form, and an oligomeric ATP-bound active form[24,25]. As shown for other NLR proteins[11,12,26], binding of ligand (maltotriose) to the sensor domain triggers a conformational change, leading to ATP/ADP exchange and oligomerization that enables MalT for transcriptional activation. Despite accumulating genetic data characterizing different MalT mutants and structural information on protein domains or on homology modeling, little is known about the structural basis of MalT activity regulation[25,27–33].

Here we report the cryogenic electron microscopy (cryo-EM) structure of an inhibitory complex consisting of MalT and MalY, which reveals a heterotetrameric assembly with a stoichiometry of 2:2 between two proteins. By analyzing different interaction interfaces contributing to MalT inhibition, we found that MalY binds to one oligomerization surface of MalT, thereby inhibiting MalT activation by blocking oligomerization. In addition, structural and biochemical data support a dual role for the loop N-terminal to the nucleotide-binding domain (NBD) in MalT autoinhibition and activation. We propose a model for maltotriose-induced MalT DNA-binding, providing insights into transcriptional regulation by prokaryotic STANDs.

## Results

### Reconstitution and cryo-EM structure of the MalT-MalY complex

To reconstitute a MalT-MalY complex, both proteins were first recombinantly expressed in *E. coli*, then purified by affinity purification and gel filtration (Fig. 1a). Since MalY is known to bind MalT in its resting form[25], ADP and PLP were supplemented during the purification steps of MalT and MalY, respectively. The elution volume of MalT corresponds to a molecular weight of 103 kDa, and MalY to 87 kDa (2 × 43.5 kDa). To assess the interaction between MalT and MalY, the purified MalT and MalY proteins were incubated in a 1:1 molar ratio and analyzed by gel filtration. The results from the assay showed the two proteins formed a stable complex with a molecular weight of about 300 kDa (Fig. 1a), consistent with a heterotetrameric assembly of the two proteins as previously reported[18].

To understand the structural mechanism of the MalT/MalY interaction, we used cryo-EM to determine the complex structure. After an initial quality check by negative staining EM, the purified MalT-MalY complex was used for cryogenic sample preparation and data collection. A total of 2802 micrographs were taken and further processed using RELION 3.1[34–36] (Supplementary Fig. 1). For 2D classification, 1,854,962 particles were automatically selected. 2D class averages showed that a dimeric MalY was bound by two MalT protomers on each side opposite to the dimer interface with C2 symmetry. The best 2D categories containing 1,145,822 particles were then used for 3D reconstruction. After 3D reconstruction and 3D classification, a subset of 176,969 particles were used for final refinement, yielding a map with a global resolution of 2.94 Å (Supplementary Fig. 1 and Supplementary Table 1).

The overall structure of the MalT-MalY complex is shaped like the capital letter "H" (Fig. 1b). In the complex, MalY forms a homodimer with each of its subunits bound by one cofactor PLP. The central portion of the dimeric PLP-binding domain forms the horizontal bar of the letter "H". The MalY dimer is nearly identical to the previously reported crystal structure[28], indicating that MalT binding induces no conformational change in MalY. The MalT structure that is well defined corresponds to the four N-terminal domains of the protein: the nucleotide-binding domain (NBD, residue 1–181), the helical domain (HD, residue 182–248), the winged-helix domain (WHD, residue 249–336), and the arm domain (residue 337–423) (Fig. 1b and Supplementary Fig. 2). Structural comparison revealed that the structure of MalT largely resembles the crystal structure of its homolog PH0952

from *Pyrococcus horikoshii* in an inactive form (Supplementary Fig. 3)[33], indicating that MalT is in an autoinhibited conformation in the cryo-EM structure. Consistently, a cryo-EM density corresponding to an ADP molecule was found between the NBD, HD, and WHD (Supplementary Fig. 4). Notably, intramolecular interactions between the NBD and WHD of MalT are directly mediated by the N-terminal segment of NBD, indicating a critical role of this segment in maintaining MalT autoinhibition (Fig. 1b). The C-terminal sensor and DNA-binding domains of MalT are not well defined in the complex structure, presumably due to their flexibility relative to the other domains of MalT.

### Structural mechanism of the MalY-MalT interaction

MalY-MalT contacts are mainly mediated by complementary charge and shape interactions (Fig. 2a). The MalT α8 helix and its N-terminal segment dominate the interactions via packing against one side of the PLP-binding domain of MalY. In addition, the α7 helix, which is closely appositioned to α8, also contacts the same side of MalY. These contacts collectively result in a buried surface of ~750 Å². MalT R173 at the C-terminal side of α8 forms a bidentate salt bond with MalY D182 and E185 and makes van der Waals packing against MalY C181. MalY C181 also establishes hydrophobic contacts with MalT V172 and N169. MalY N219 tightly stacks against the Cα atom of G166 and the side chain of A168 from MalT. Further at the N-terminal side of α8, the three consecutive MalT residues, L162, P163 and Q164, are buttressed by MalY A84. MalT R143 and R147 dominate the interaction of α7 with MalY. R143 makes a hydrogen bond with the carbonyl oxygen atom of MalY A221 and van der Waals interactions with the carbonyl oxygen atoms of MalY S218 and N219, whereas R147 contacts MalY E185 and R222 via van der Waals interactions. Interestingly, MalT R171, a residue that has been reported to mediate both ATP hydrolysis and oligomerization[31], is also closely located to the MalY-recognizing surface. Though R171 does not contact MalY directly, it forms a hydrogen bond with E178 and might stabilize the C-terminal segment of MalT α8 helix.

In support of our structural observation, several MalY residues including A84, C181, E185, S218, N219, and A221 from the MalY-MalT interface have been mapped as functionally important by previous mutagenesis screens[28]. Consistently, the MalY from *Vibrio furnissii* whose D182 and S218 are substituted by cysteine and valine, respectively, is unable to repress a MalT-dependent gene despite high sequence homology[22,37]. To further verify our structure, we mutated MalT residues from the MalT-MalY interface in the cryo-EM structure and assessed the interaction of the resulting mutants with MalY using pull-down assays (Fig. 2b). As anticipated, the MalT R143A and R173A mutations at the center of the interface completely abolished interaction with MalY in the assays. In contrast, the MalT E140A and N169A mutations had little effect on the interaction.

The effect of NBD mutations on MalT inhibition by MalY was further investigated in vivo using *malY⁺* and *ΔmalY* derivatives (strains G and H, respectively) of a reporter strain harboring the *lacZ* gene (which encodes the β-galactosidase) under the control of a MalT-dependent promoter. MalT was expressed at a low level from a single-copy plasmid (pJB215). Both strains are devoid of *malK* and *aes*. As a result, in the *ΔmalY* reporter strain, WT MalT is activated by endogenous maltotriose while its activity is clearly diminished in the *malY⁺* reporter strain (Fig. 2c). In contrast, all three MalT mutants that lost MalY interaction in vitro (MalT-R143A, MalT-R173A, and MalT-R143A/R173A) showed comparable β-galactosidase activities in the two strains. These results suggest MalY-mediated inhibition of MalT is relieved by these interface mutations. Furthermore, the transcriptional activity of the MalT double mutant was strongly reduced in comparison to that of WT, suggesting a role of these two residues in MalT function. Taken together, the biochemical and functional data confirm the MalT-MalY interaction observed in the cryo-EM structure.

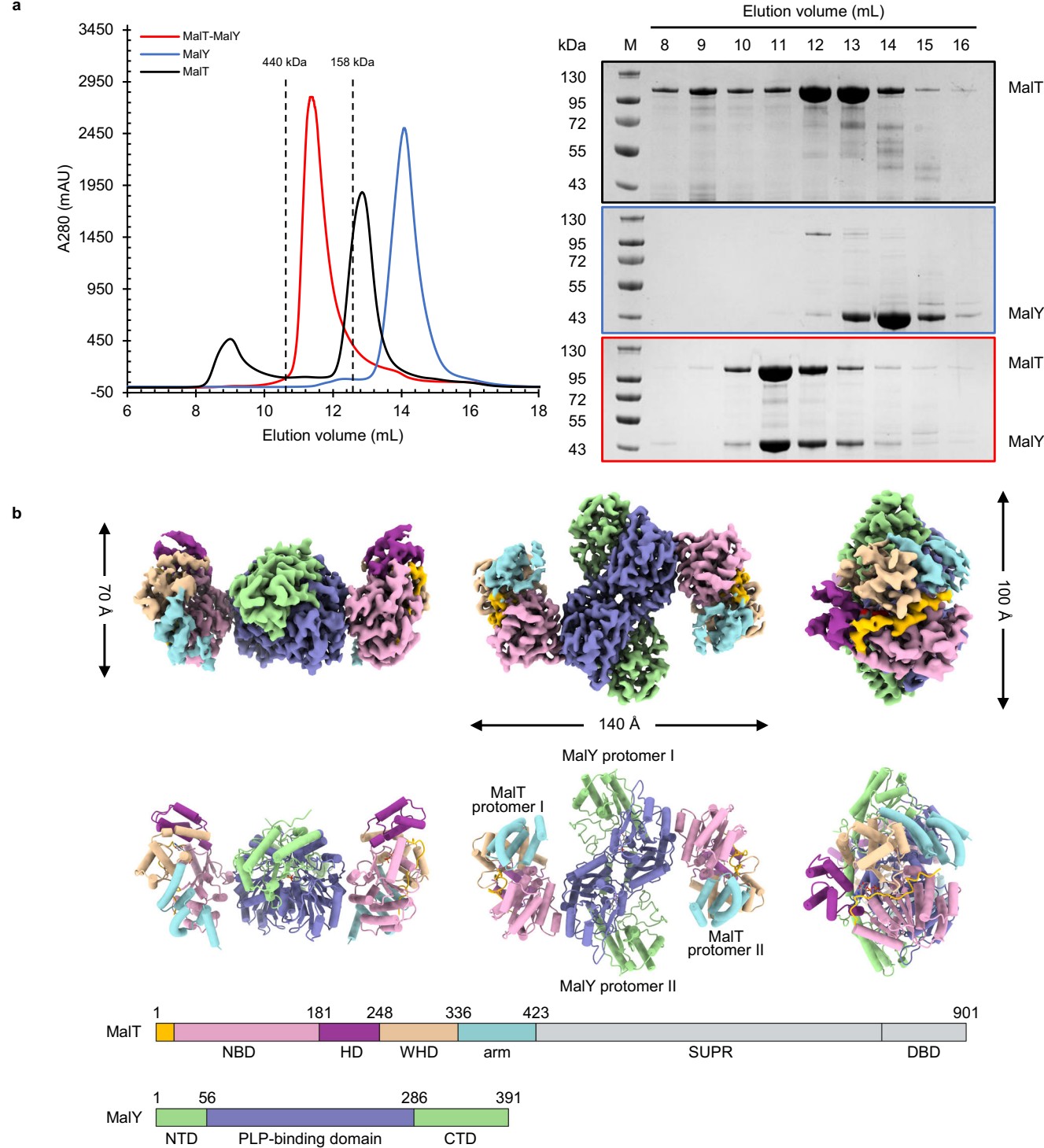

**Fig. 1 | Reconstitution and cryo-EM structure of the MalT-MalY complex.**
**a** Affinity-purified MalT and MalY proteins were further analyzed by gel fil-
tration using a Superdex 200 Increase 10/300 GL column in the presence of
ADP (left). When MalT and MalY were incubated in a 1:1 molar ratio, the two
proteins formed a stable complex with a molecular weight of about 300 kDa.
The 9-mL peak observed when MalT is filtered alone may contain protein
aggregates or MalT protein that partially oligomerized at a high protein
concentration (>3 mg/ml). The presence of MalT and MalY in fractionated
samples was confirmed by SDS-PAGE analyses (right). The experiments have
been repeated for three times with similar results. **b** Cryo-EM density map
with 2.94 Å resolution (top) and the refined structure model (middle) of the
MalT-MalY complex (PDB: 8BOB) shown in different orientations. Each MalT
or MalY protomer is labelled. Residue number and colors of each protein
domain are indicated (bottom). The N-terminal segment of NBD is high-
lighted in yellow. The C-terminal domains of MalT are not well defined in the
final structure. DBD DNA-binding domain.

## MalY represses MalT activity by blocking oligomerization

To explore the mechanism of MalT inhibition by MalY, we first made
structural alignments between MalT-MalY and other known protein
structures from the STAND family (Fig. 3a, b). When the NOD module
of MalT was used as a query protein structure, the NOD of Apaf-1 was
identified as one of the closest homologs by a DALI search. Super-
imposition of the NBD and HD of MalT from the complex structure
with one protomer in an Apaf-1 lateral dimer of the Apaf-1 apoptosome

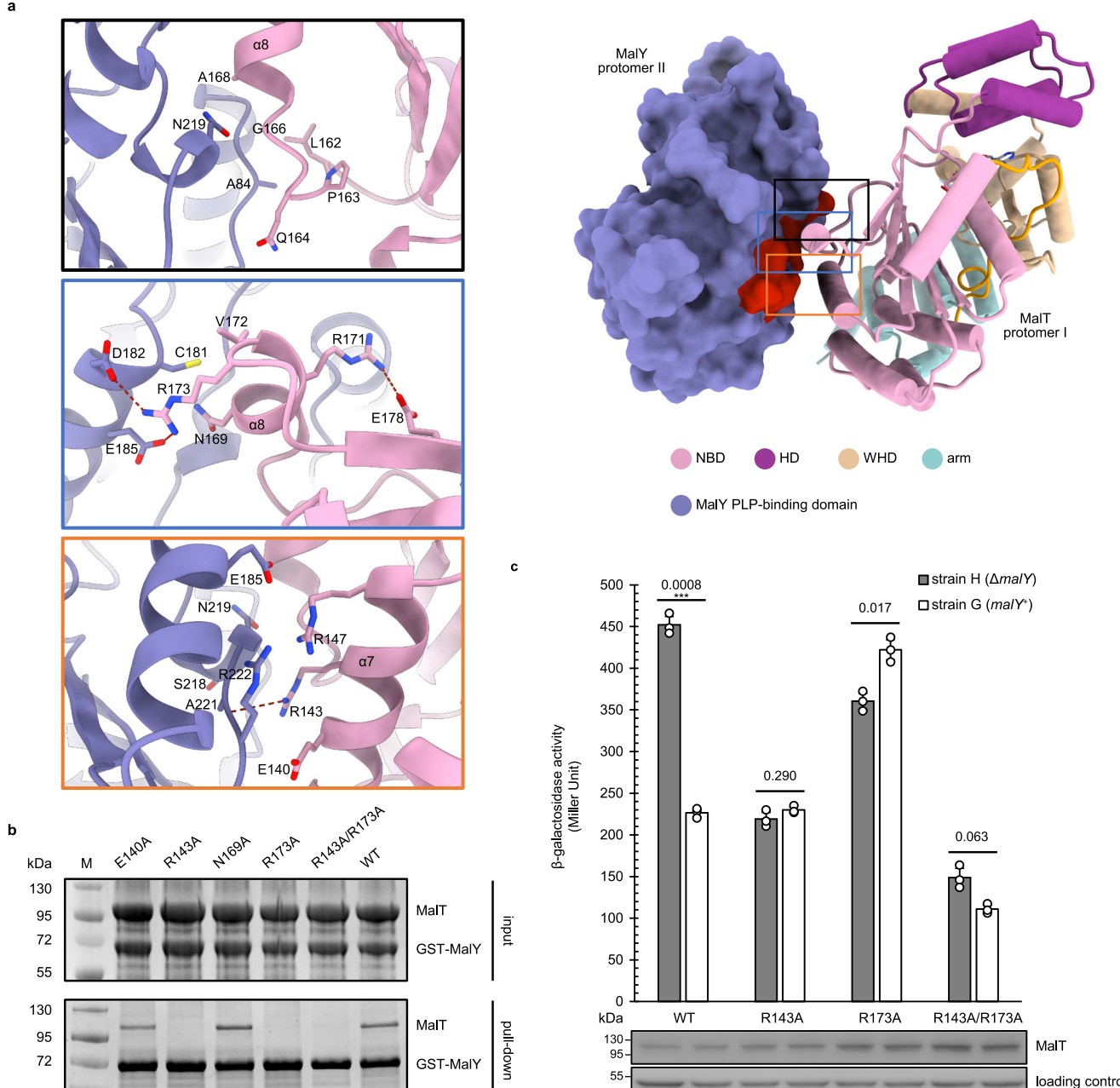

**Fig. 2 | In vitro and in vivo analyses of MalY-MalT interaction. a** The interface between MalT and the PLP-binding domain of MalY with structural details. The MalT-recognizing surface patch on MalY is highlighted in red (right), polar interactions are represented by dashed lines, residues involved in hydrophobic packing are also shown (left). Each MalT or MalY protomer is labelled, and colors of each protein domain are indicated. The N-terminal segment of NBD is highlighted in yellow. **b** Pull-down assay using GST-tagged MalY protein and WT or mutant MalT proteins carrying interface mutations. The experiments have been repeated for three times with similar results. **c** Levels of β-galactosidase activity in strains G and H harboring WT MalT plasmid (pJB215) or a derivative thereof and grown in a minimal medium supplemented with glycerol. The enzymatic activity values obtained were corrected for the background as measured with strains harboring empty vector (pJM241). The values given are the means ± SD of results from three independent experiments. The asterisks indicate significance of two-tailed Student's *t*-tests, ***$P < 0.001$. MalT proteins were detected by western blot using total-cell extracts from the assayed cultures. A nonspecific band with lower molecular weight that appeared in all the samples was used as loading control.

revealed that MalY almost completely overlapped with the other Apaf-1 protomer[8]. Similar results were also found when comparing MalT-MalY to a lateral dimer of CED-4 or ZAR1 in their active forms[11,38]. Because conserved surfaces are involved in NOD-mediated oligomerization of STAND proteins[39], these structural observations suggest that MalY binding masks one oligomerization surface on MalT and consequently inhibits MalT. Structural alignment of MalT-MalY with an inactive NLRC4 molecule revealed that MalY and the LRR of NLRC4 are similarly positioned, suggesting a functional similarity between MalY and the LRR of NLRC4 in sequestering MalT and NLRC4, respectively, in a resting state[40].

We next investigated if the MalY-interacting surface is involved in MalT oligomerization. We therefore analyzed the self-association capability of the MalT NBD mutants that lost MalY-binding activity by gel filtration (Fig. 3c). In the presence of maltotriose and ATP, the sizes of the oligomers formed by MalT-R143A and MalT-R173A were notably reduced compared to those of WT. The double mutant carrying both substitutions was eluted at a position close to that of monomeric MalT,

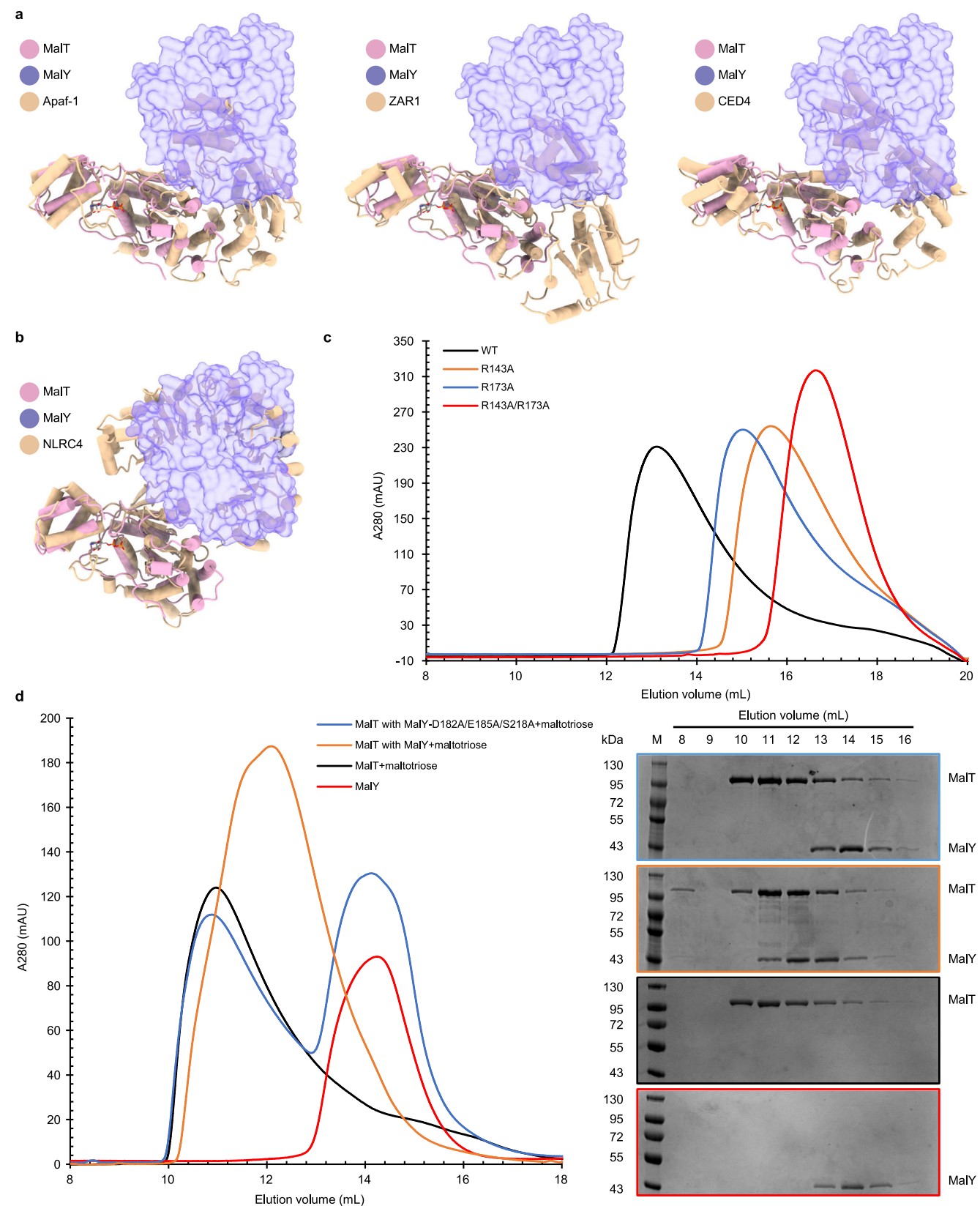

further supporting a role of these two residues in MalT oligomerization. By contrast, oligomerization of MalT-E140A and MalT-N169A, two mutants that could still interact with MalY, was not compromised (Supplementary Fig. 5). Altogether, these results confirmed a critical role for residue R143 and R173 in mediating MalY recognition and MalT oligomerization, suggesting that MalT oligomerization and interaction

with MalY are exclusive. Indeed, incubation of MalT and MalY in the presence of 1 mM maltotriose and 0.4 mM ATP, i.e., under MalT oligomerization conditions, substantially compromised the MalT-MalY interaction as indicated by gel filtration[18]. The data above thus suggests that MalY interferes with MalT activation by inhibiting its oligomerization. To test this idea, we assessed the impact of MalY

**Fig. 3 | MalY represses MalT activity by blocking oligomerization.** The structure of MalT NBD-HD and one interacting MalY molecule (shown in transparent surface view) from the MalT-MalY complex was aligned to (**a**), lateral dimers comprising the NBD and HD of Apaf-1, ZAR1, and CED-4, or (**b**), an inactive NLRC4 consisting of its NBD, HD1, and LRR. **c** A same molar amount of MalT WT, R143A, R173A, and R143A/R173A proteins were pre-incubated and subjected to gel filtration analyses using a Superose 6 Increase 10/300 GL column in the presence of 1 mM maltotriose and

0.4 mM ATP. **d** A same molar amount of MalT and MalY WT or MalY-D182A/E185A/S218A proteins were preincubated and subjected to gel filtration analyses using a Superdex 200 Increase 10/300 GL column in the presence of 1 mM maltotriose and 0.1 mM ADP, either separately or together (left). The presence of MalT and MalY in fractionated samples was confirmed by SDS-PAGE analyses (right). The experiments have been repeated for three times with similar results.

mutations altering the MalT-MalY interface on MalT oligomerization (Fig. 3d). Gel filtration was performed in the presence of ADP and maltotriose. The inducer is indeed known to trigger MalT oligomerization in the presence of ADP, albeit with a reduced efficiency compared to what is observed in the presence of maltotriose and ATP[24]. MalT incubation with MalY greatly shifted MalT to lower molecular weight species, indicating that MalY blocks MalT oligomerization. This agrees with a previous report that ADP-bound MalT is much less sensitive to maltotriose when complexed with MalY[18]. The MalY triple mutant (MalY-D182A/E185A/S218A) that carries interface mutations had little effect on MalT oligomerization induced by maltotriose and ADP, indicating that MalY binding is important for the inhibition. Taken together, these results suggest that MalY represses MalT activity by blocking its oligomerization.

### N-terminal loop (N-loop)-mediated autoinhibition of MalT

In Apaf-1 and its *Drosophila* homolog Dark, the ligand-binding WD40-repeats domain serves as an anti-apoptotic factor in the absence of stimuli[41–43]. Likewise, in NLRC4 and ZAR1, the LRR sequesters the NBD, which stabilizes the resting form[40,44]. A marginal contact between NBD and the sensor domain of MalT has been proposed to strengthen MalT inhibition by facilitating MalK binding[33]. However, this contact was not observed in our structure as the C-terminal domains of MalT were not well defined, suggesting that such a contact, if it does exist, is dispensable for MalT autoinhibition in the absence of MalK. An important feature of MalT is that the effector domain is not at the N-terminal end but is connected to the C-terminal sensor domain. The first 20 amino acids N-terminal to the NBD form a loop structure (hereafter called N-loop) wedged between the WHD, HD, and NBD (Fig. 4a). A similar loop segment is also present in the structure of the inactive PH0952, but forms much fewer contacts with other structural domains (Supplementary Fig. 3)[33]. In all the other known inactive NOD structures, ADP-mediated NBD/WHD interaction plays a pivotal role in autoinhibition[33,40,44,45], in particular via a hydrogen bond between the β-phosphate and a conserved WHD histidine. This interaction is also conserved in the structure of the resting MalT (Supplementary Fig. 4). In addition, the NOD module of MalT is kept in a closed form by direct interactions of the N-loop with NBD and WHD. Therefore, disruption of these interactions should promote NOD opening and MalT activation. Consistently, gain-of-function mutations in the N-terminal region have been isolated by genetic approaches (MalT-S5L, MalT-R9S), which result in a higher transcriptional activity compared to that of WT[30]. Furthermore, substitution of residues M311 or W317 from WHD enhances MalT activity[30,32]. Though the side chains of these two residues form no contacts with the N-loop, their substitutions could destabilize NBD/WHD interactions by altering the conformation of the WHD surface.

Using structure-guided mutagenesis, residues of the N-loop that are involved in polar interactions with NBD and WHD were systematically mutated to alanine. The effects of these mutations as well as the gain-of-function mutations aforementioned were evaluated in vivo in the absence of any MalT inhibitory protein by measuring the levels of β-galactosidase activity in strain H expressing MalT from plasmid pJB215 as above (Fig. 4b). In support of previous data[30], the S5L and R9S mutations strikingly promoted the transcriptional activity of MalT. Similarly, the mutations of S8A and T16A caused 8.5-fold and 2.5-fold enhancement of MalT transcriptional activity, respectively. As a negative control, mutation of R12 which is not involved in interaction

with NBD or WHD had no detectable effect on the MalT activity. It should be noted that, as previously observed for gain-of-function variants[32,33], these mutants with enhanced transcriptional activity displayed higher protein levels compared to WT MalT. This recurrent feature suggests that the active form of the protein is more resistant to proteolysis than the resting form. Nonetheless, the transcriptional activity enhancement of S8A and R9S was more striking than the increases in their protein levels. Thus, these two mutations had a more pronounced effect than the others on MalT activation.

As shown for other STANDs[7,8,11], nucleotide exchange is an essential step of the MalT activation process. If N-loop mutations facilitate MalT activation by removing its contacts with NBD and WHD, the autoactive mutants are likely to be prone to ATP/ADP exchange in the absence of inducer. To investigate this hypothesis in vitro, monomeric WT or mutant proteins were first purified in the presence of ATP at a saturated concentration. After removing free nucleotides from the samples by gel filtration, the protein-bound nucleotides were released, and their amounts quantified by a luciferase assay (Fig. 4c). While WT MalT protein contained only ADP as previously reported[25], both ATP and ADP were detected for MalT-S5L and MalT-R9S, suggesting that these mutations promote exchange of ADP for ATP. These results are consistent with those of the in vivo transcriptional activity assays. Unexpectedly, nucleotide exchange was not observed for MalT-S8A, although this substitution had a strong effect on MalT activity. These results suggest that the effect of this mutation on NOD opening may not be tracked by this assay. Whereas substitution T16A causes a slight increase in MalT activity, its effect on nucleotide exchange might be too small or not detectable by this assay either. As expected, R12A did not affect nucleotide binding of MalT.

### MalT N-terminal region contributes to protein oligomerization

Given the repressive function of the N-loop, it is intuitive to assume that its truncation would lead to MalT activation. Surprisingly, however, deletion of the first 8 amino acids completely abolished MalT transcriptional activity and reduced protein stability in vivo (Supplementary Fig. 6), suggesting that the N-loop also has a role in MalT activation. Consistently, the K6A and R19A mutations, which alter two other N-loop residues that form polar contacts with NBD (Fig. 4a), resulted in a loss of the transcriptional activity of MalT (Fig. 5a). To further test this model, oligomerization of all the N-loop mutants was examined by gel filtration in the presence of maltotriose and ATP. All of them exhibited a lower degree of oligomerization compared to the WT protein, suggesting that the N-loop mediates MalT oligomerization (Fig. 5b). K6A and R19A appeared to have more striking effects on MalT oligomerization because the two mutant proteins were monomeric in the assay, supporting a critical role for the mutated residues in MalT activation. The discrepancy between the enhanced in vivo transcriptional activities of the variants and their reduced oligomerization in vitro (e.g., S5L, S8A, and R9S) is only apparent. The oligomerization assay monitors their intrinsic ability to self-associate, which is insensitive to NOD destabilization caused by N-loop substitutions in the presence of ATP and maltotriose. In contrast, since the transcriptional activity of MalT is rate-limited by NOD opening in vivo[32], compromised autoinhibition results in an increased level of activated MalT protomers that more than compensates an oligomerization defect.

Analysis of the nucleotide content of MalT-K6A revealed that the variant is predominantly ATP-bound (Fig. 5c), indicating K6 also plays a

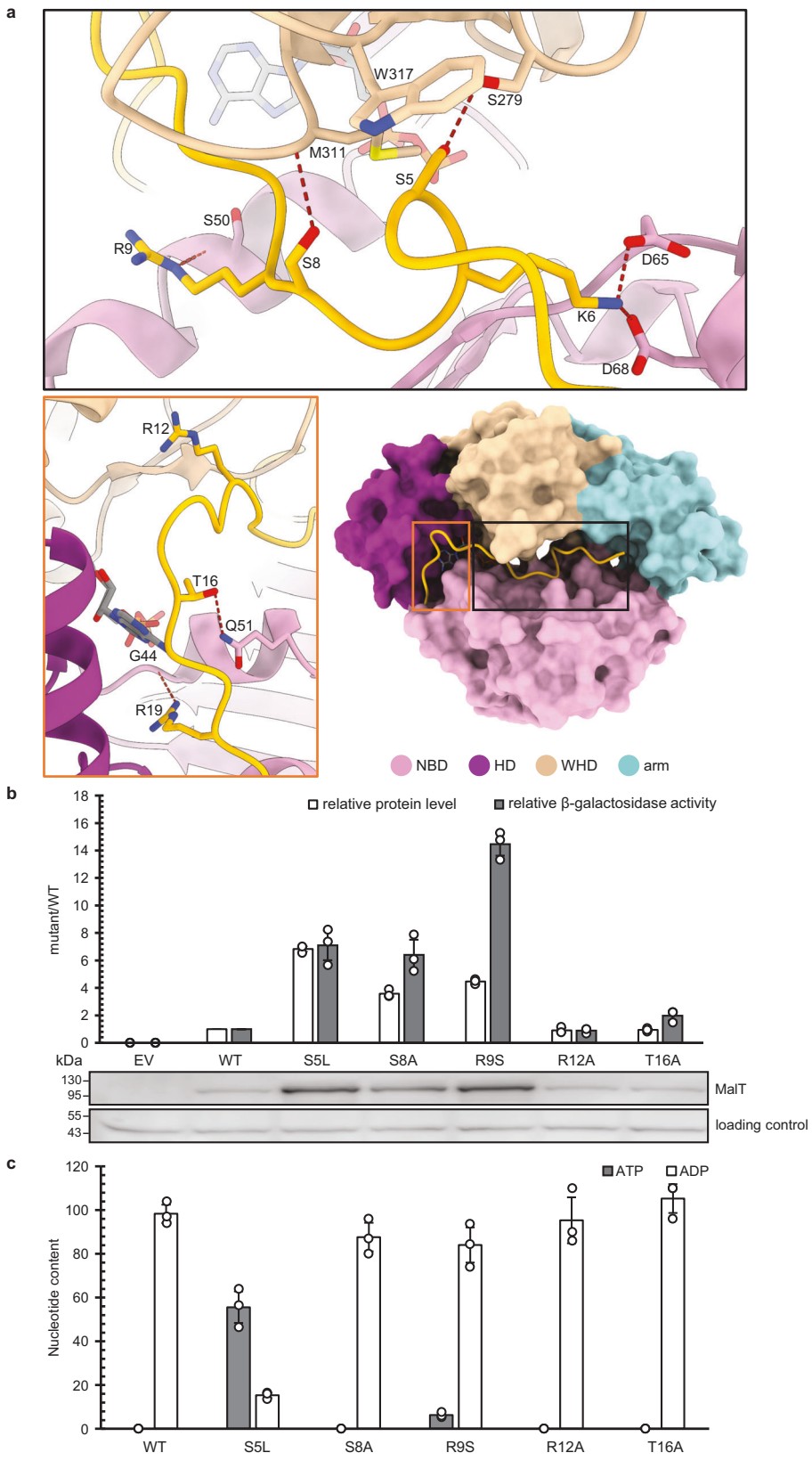

role in MalT autoinhibition. This conclusion is further supported by the gain-of-function phenotype conferred by substitution of the NBD residues hydrogen-bonded to K6 (D65 and D68)[32], likely by destabilizing the contact between N-loop and NBD. Likewise, detection of ATP bound to MalT-R19A reveals a role of R19 in autoinhibition, but it might

be indirect given the reduced nucleotide-binding activity (Fig. 5c). Interestingly, in the structures of active ZAR1, Sr35, Apaf-1, and NLRC4, a loop or helix N-terminal to the NBD is wedged between two protomers and mediates their oligomerization. Residues from the N-terminal regions of these proteins are also involved in ATP binding.

**Fig. 4 | N-terminal loop (N-loop)-mediated autoinhibition of MalT. a** The first 20 amino acids N-terminal to the NBD of MalT form a loop structure (highlighted in yellow) and make extensive contacts with both NBD and WHD. Polar interactions are represented by dashed lines, colors of each protein domain are indicated. **b** In vivo activities of MalT WT and N-loop mutants. The levels of β-galactosidase activities were determined by using strain H harboring pJB215 or a derivative thereof and corrected for the background as described in Fig. 2. The values given are the ratios of the mutant activities to that of WT. The relative protein levels of MalT mutant to that of WT were determined by western blot quantification using total-cell extracts from the assayed cultures. A non-specific band with lower molecular weight that appeared in all the samples was used as loading control. All the values are the means ± SD of results from three independent experiments. **c** Luciferase assays of MalT WT and N-loop mutants. Proteins were first purified in the presence of 0.4 mM ATP. Free nucleotides were then removed from the samples by gel filtration, the protein-bound nucleotides were released, and their amounts quantified. The nucleotide content was calculated as molar percentage of nucleotide to protein. All the values are the means ± SD of results from three independent experiments.

These results confirm that the region N-terminal to the NBD has a conserved role in both ATP binding and oligomerization of STAND proteins as noted before[39].

## Activation of MalT stabilizes its C-terminal domain for DNA-binding

Oligomerization is important for MalT DNA-binding activity[46]. To gain insights into the underlying mechanism, oligomeric protein samples were prepared for cryo-EM analysis. In the presence of maltotriose and ATP, MalT formed open ring-like structures with different sizes (Fig. 6a), in agreement with a previous study[47]. Due to preferred orientations of the MalT particles, we have been unsuccessful in reconstructing their 3D structures. Nevertheless, the 2D class averages of these particles revealed more structural information than those previously reported, and a low-resolution 3D map was reconstructed (Supplementary Fig. 7a). Given that DNA interacts with the outer layer of MalT oligomers[47], the sensor domain, and DNA-binding domain should be located in the peripheral region of the open ring-like structures, where several patches of densities are arranged in apposition (Fig. 6a). An Alphafold2[48]-predicted monomeric MalT model in active conformation could be well fitted into the 3D map of oligomers (Supplementary Fig. 7b), suggesting the patches of density likely correspond to the C-terminal domains. These structural domains of MalT are flexible when complexed with MalY, but are stabilized upon activation. Such a change is likely a prerequisite for promoter binding and activation of transcription initiation. In addition to oligomerization, the binding of maltotriose may have a structural role in stabilizing the C-terminal domains, since MalT-W317R, a MalT mutant that can fully oligomerize in the absence of inducer, still requires maltotriose to bind the promoter MalT sites[32].

To further assess the relationship between MalT oligomerization and promoter-binding, we investigated how the N-loop mutations impact the DNA-binding activity of MalT. An electrophoretic mobility shift assay (EMSA) was used to detect WT or mutant MalT proteins interaction with a DNA fragment containing *malPp800*, a MalT-dependent promoter harboring four MalT binding sites (Supplementary Fig. 8). All the N-loop mutants except MalT-K6A and MalT-R19A retained DNA-binding capability, which is consistent with the inability of MalT-K6A and MalT-R19A to oligomerize. Except for MalT-S5L, all the variants form a complex with the promoter DNA similar to that formed by WT MalT despite of their reduced ability to self-associate, which is essential for transcriptional activation by MalT.

## Ligand-induced MalT activation

As observed for other STANDs, ligand-induced exchange of ADP with ATP is required for the activation of MalT[25,44,45]. Furthermore, the structures of the NOD module are highly conserved between MalT and other STANDs (Supplementary Fig. 9). These results suggest that they may share a similar mechanism for ligand-induced nucleotide exchange during activation. However, our cryo-EM structure of inactive MalT does not provide any clue about the maltotriose binding site because of the flexibility of the C-terminal sensor and DNA-binding domains. A prediction of inactive MalT structure from AlphaFold2[48]

largely resembles the cryo-EM structure of MalT for the NOD module, suggesting robustness of the prediction (Supplementary Fig. 10a). In the predicted structure, the arm and sensor domains are located adjacent to the NBD, and loose packing of these three structural domains form a large groove. A previous study showed that the NBD and arm domains of inactive MalT are in close vicinity, and the recognition of maltotriose begins with a low-affinity binding step involving solely the sensor domain, which is followed by recruitment of the arm to form a high-affinity binding site[49]. These results suggest that the groove might be critical for maltotriose binding to MalT. Consistent with this model, double mutations of S383R/N386R or W524R/I532R considerably decreased the transcriptional activity, and the activity of a quadruple mutant carrying these four substitutions was reduced to <20% of WT MalT activity. In the meantime, mutation of residues localized further away from the groove had little effect (Q410R, S421Y, F555R) (Supplementary Fig. 10b). While the C-terminal domains of MalT are flexible in resting form, maltotriose binding may stabilize the conformation of the sensor domain, which could consequently clash with NBD and trigger structural remodeling of NOD that allows exchange of ADP for ATP, as what has been shown for ZAR1, Sr35 and RPP1[11,12,26]. Supporting this hypothesis, the hinge connecting the arm and the sensor of MalT is only cleaved by proteinase K when the arm-sensor module is unliganded[49,50]. It is of note that several residues located in this groove were also identified to be important for maltotriose binding by a different method[51]. A thorough test of this model by future structural and biochemical investigations is needed.

## Discussion

MalT is a prokaryotic STAND that has a unique structural organization compared to canonical STANDs because the effector domain is C-terminally positioned. Nonetheless, our cryo-EM analysis showed that the structure of the NOD module of MalT is highly conserved compared to the other inactive STAND proteins (Supplementary Fig. 9), indicating that MalT shares the same mechanism of NOD autoinhibition. Indeed, the closed state of the NOD module is stabilized by the N-loop sandwiched between NBD and WHD, which glues these two domains together. Supporting a role of the N-loop in MalT autoinhibition, mutations of some residues in this region or the N-loop-contacting regions of NBD/WHD promoted MalT transcriptional activity. In contrast to the inactive NLRs and Apaf-1 with structures known, however, the C-terminal sensor and DNA-binding domains are invisible in the inactive form of MalT, suggesting that these two domains are not involved in autoinhibition when MalT is complexed with MalY. Interestingly, structural comparison revealed that MalY is similarly positioned as the C-terminal LRR domain of NLRC4, suggesting that MalY serves a similar function as the LRR domain of NLRC4 in inhibiting MalT activation. Indeed, as demonstrated here, MalY binding strongly suppresses MalT oligomerization (Fig. 3d) and the MalY-recognizing patch overlaps one MalT oligomerization surface. MalY inhibition of MalT is reminiscent of CED-9 mediated inhibition of CED-4, in which CED-9 sequesters the CED-4 dimer from further oligomerization by contacting one oligomerization surface of CED-4[52]. Unlike CED-4, the ADP-bound NOD module of MalT exists in an autoinhibited conformation, and MalY offers additional control

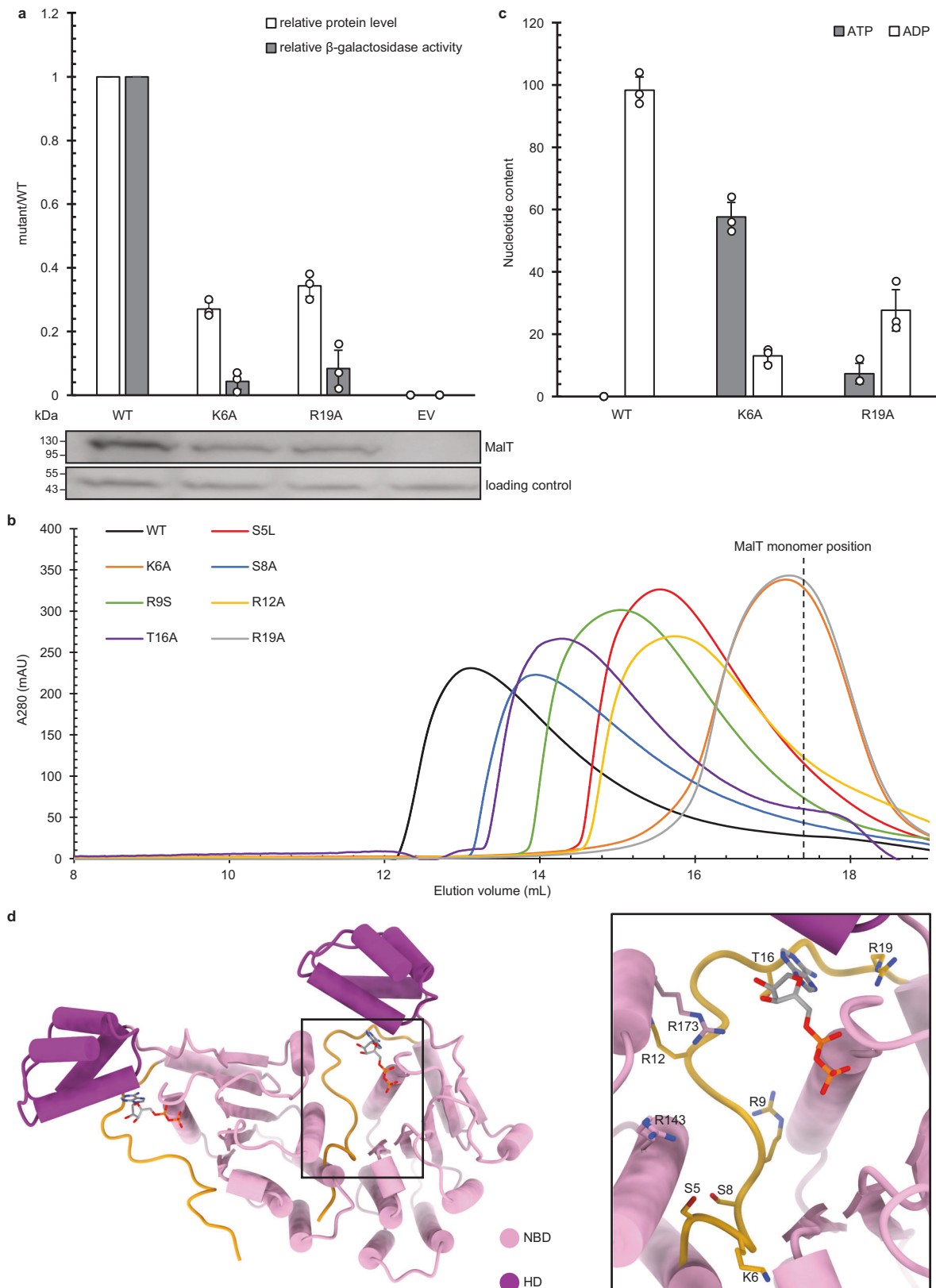

over MalT activation by inducer molecules present around. However, the step targeted by MalY during MalT oligomerization is enigmatic. MalY might sequester a pre-activated MalT molecule with a liganded sensor domain[49] that would otherwise be recruited by an activated MalT molecule. This recruitment process, in addition to self-

association of fully activated protomers, may also contributes to MalT activation.

A repressive function is shared by the N-terminal regions of different NLRs. Like the N-loop of MalT, the N-terminal CARD contributes to Apaf-1 autoinhibition by stacking closely against

**Fig. 5 | MalT N-terminal region contributes to protein oligomerization. a** In vivo activities of WT MalT or N-loop mutants that abolished protein oligomerization. β-galactosidase activities were determined by using strain H harboring pJB215 or a derivative thereof and corrected for the background as described in Fig. 2. The values given are the ratios of the mutant activities to that of WT. The relative protein levels of MalT mutant to that of WT were determined by western blot quantification using total-cell extracts from the assayed cultures. A non-specific band with lower molecular weight that appeared in all the samples was used as loading control. All the values are the means ± SD of results from three independent experiments. **b** A same molar amount of MalT WT or N-loop mutant proteins were pre-incubated and subjected to gel filtration analyses using a Superose 6 Increase 10/300 GL column in the presence of 1 mM maltotriose and 0.4 mM ATP. **c** Luciferase assays of WT MalT or N-loop mutants that abolished protein oligomerization. The assays were done as described in Fig. 4. All the values are the means ± SD of results from three independent experiments. **d** A modeled MalT dimer containing NBD and HD. The N-loop is highlighted in yellow. The structure was modeled based on an Apaf-1 lateral dimer from the Apaf-1 apoptosome. Both the N-terminal loop and MalY-interacting interface of MalT are in the dimerization surface in the modeled structure. Residues contributing to protein oligomerization are shown.

NBD and WHD and thereby burying the procaspase-9-binding surface[45]. Interestingly, N-terminal K6A and R19A mutations resulted in a complete loss of the transcriptional activity of MalT, suggesting that the N-loop is also important for MalT activation. This model is consistent with the cryo-EM structures of several STAND proteins including ZAR1, NLRC4, and Apaf-1 in their active states[7,8,11]. In these structures, a loop or helix N-terminal to the NBD mediates inter-protomer interaction, a role that has been confirmed for ZAR1 by biochemical and functional data[11]. These results therefore suggest that the N-terminal region of the NBD has a conserved function in mediating oligomerization of a STAND protein. Biochemical data showed that the mechanisms whereby the K6A and R19A mutations abolish MalT activity are different. Based on the modeled structure of a MalT NBD-HD dimer (Fig. 5d), K6 is situated at the dimer interface and likely mediates MalT oligomerization through direct contact with the neighboring protomer. In the inactive MalT structure, R19 hydrogen bonds with G44, a P-loop residue interacting with the β-phosphate group of the MalT-bound ADP (Fig. 4a and Supplementary Fig. 4). Distinct from K6, whose substitution does not abrogate ATP-binding, substitution of R19 could affect the conformation of the P-loop, thereby reducing the nucleotide binding affinity of MalT and disrupting ATP-mediated inter-protomer interactions. Notably, the arginine present in the N-terminal helix of P-loop is a conserved feature shared by STAND proteins[1]. In the inactive Apaf-1 structure, R129 is also found to stabilize the conformation of P-loop by making a hydrogen bond with C158[45].

Combining the results from this and previous studies, a mechanistic model regarding the activation of MalT can be deduced (Fig. 6b). When no substrate is transported through the maltose transporter, a fraction of MalT protein is anchored to the membrane via interactions between its NBD and sensor domains and the MalK subunit of the transporter. In this conformation, the sensor domain of MalT could be stabilized by interacting with both NBD and MalK[33], and the maltotriose-binding site may be inaccessible. Meanwhile, MalT can also be sequestered in the cytoplasm by cytosolic inhibitory proteins, such as MalY or Aes, which prevents its accidental activation by endogenously synthesized inducer. MalT binding by MalK, MalY, and Aes is not likely to occur simultaneously, since the repressions of MalT by MalK and Aes are both mediated through residue T38 from MalT (Supplementary Fig. 11)[19,20]. Moreover, MalK and Aes will clash with MalY if they bind MalT through this residue on NBD (Supplementary Fig. 11b, c). The independent function of repressor proteins could incorporate information from different signaling pathways, with MalT being a nexus that modulates the activity of the maltose system. When MalT is complexed with MalY, the oligomerization surfaces of NBD are both masked due to MalY sequestration and the autoinhibition mediated by the N-loop. Assuming the WHD of MalT and Apaf-1 moves similarly relative to NBD-HD during activation[8], binding of maltotriose to the sensor and arm domains of MalT triggers conformational changes in the NOD that can allosterically disengage the inhibitory contacts (Supplementary Fig. 12), rendering MalT oligomerization-competent. This results in stabilization of the DNA-binding domains for cooperative interactions with the MalT-binding sites in the target promoters[49]. In this sense, the oligomerized MalT is like the Apaf-1 apoptosome and NLR inflammasomes that can recruit downstream signaling components, but different from plant NLR resistosomes that function as either channel, NADase or holoenzymes. The MalT-MalY complex structure here provides insights into the inhibition mechanism of a non-canonical STAND by a repressor protein, and sheds light on the mechanism whereby the activity of a STAND transcription factor is controlled.

## Methods

### Strain and plasmids

ED169 (MC4100 Δ*malB107*)[53] was used as the starting strain for constructing strain G (MC4100 Δ*malB107 trp*::[Kan$^r$-*malEpΔ92-lac*]$_{op}$ Δ*malT220* ΔcsgA::*aadA7* Δ*aes* (F$^+$)) and strain H (strain G Δ*malY*::Zeo$^r$) used for β-galactosidase assays. Δ*malB107* deletes most of the *malK* gene[54]. The transcriptional *trp*::[Kan$^r$-*malEpΔ92-lac*]$_{op}$ fusion was brought into ED169 by P1 transduction from pop4146 (MC4100 *aroB glpD* Δ*malEpKp99*::[Ptac-Pcon] *trp*::[Kan$^r$-*malEpΔ92-lac*]$_{op}$ (F$^+$)) (lab collection), generating strain A (ED169 *trp*::[Kan$^r$-*malEpΔ92-lac*]$_{op}$). *csgA* gene was deleted by P1 transduction from UGB935 (MG1655 ΔcsgA::*aadA7*) (obtained from C. Beloin, Institut Pasteur). The F factor was introduced into strain A for M13 phage infection by conjugation with pop7128 (MC4100 *malTp10* (F$^+$)) (lab collection). Deletions of *malT*, *aes*, and *malY* were carried out in a sequential order by allelic exchange using M13 bacteriophage derivatives as shuttles[46], and the deletions were verified by PCR with primers flanking the deleted region.

Phage M13mp11::Δ*malY*::Zeo$^{r32}$ was used for deleting *malY*. The *malT* gene was deleted by using the M13Zeo2::Δ*malT220*, constructed as follows. The fragment extending from positions -14 bp to +3100 bp relative to the *malT* transcription start site was amplified by PCR from BL21 (DE3) Δ*malT220* with the introduction of a HindIII site and a BglII site upstream and downstream of the homologous region, respectively. After restriction enzyme digestion, the resulting fragment was ligated between the HindIII and BamHI sites of the zeocin-resistant M13Zeo2 vector. M13Zeo2::Δ*aes* used for the *aes* gene deletion was obtained in a similar way, except that the flanking regions of *aes* (−327 bp to +3 bp and +939 bp to +1269 bp, respectively) were amplified separately and co-ligated between the HindIII and BamHI sites of the M13Zeo2 vector.

pJM241 is a single-copy R1 runaway plasmid[55]. pJB215 is modified from pOM215, a pJM241 plasmid containing P$_{KAB-TTCT}$-*malT*[21]. pJB215 bears a 57 bp insertion between codons S121 and P122 of the *malT* gene, which encodes a 2×HA tag for western blot detection. The tag is localized in a loop region of the NBD and does not interfere with MalT function based on our cryo-EM structure. pOM206 is a pET24a (+) (Merck-Millipore) derivative encoding an N-terminally His-tagged MalT[25]. Sequence of full-length *malY* was amplified from *E. coli* genomic DNA with the introduction of a NdeI site and a XhoI site at the 5' and 3' ends of the gene, respectively, digested by NdeI and XhoI and ligated between the corresponding sites of the pGEX-6P-1 vector. All the *malT* and *malY* variants used for recombinant protein expression

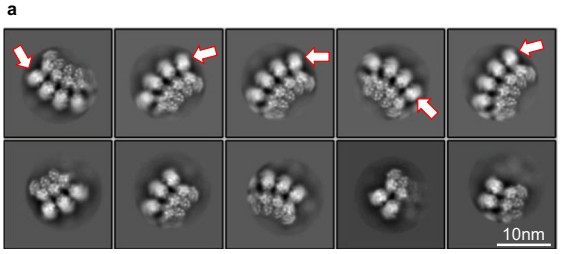

**Fig. 6 | Activation of MalT stabilizes its C-terminal domain for DNA-binding.**
**a** 2D classification from cryo-EM analyses of active MalT. The density of C-terminal domains is indicated by red empty arrows. Particles show a preferred orientation. **b** A model for MalT regulation. In the absence of substrate transport, the idling *E. coli* maltose transporter MalFGK$_2$ anchors MalT to the cytoplasmic membrane via MalK-mediated interactions. Meanwhile, MalT molecules that remain in the cytoplasm are sequestered by inhibitory proteins. Sugar transport triggers a conformational change in the MalK dimer which frees membrane-localized MalT into cytoplasm[21,67]. The inhibitory effect of MalY can be relieved by an increased concentration of maltotriose. Binding of inducer to the sensor domain of MalT (1) is followed by a high-affinity binding step involving both the sensor and arm domains[49], driving the MalT activation pathway towards dissociation of the MalT-MalY complex (2), opening of NOD (3) and oligomerization. Maltotriose binding and oligomerization together stabilize the C-terminal domains of MalT, allowing the activator binding to promoter DNA and RNA polymerase (RNAP) recruitment for subsequent transcription initiation. The NBD, HD, WHD, arm, sensor, and DNA-binding domains of MalT are colored in pink, purple, wheat, cyan, slate, and green, respectively. Domains are colored in gray if they are flexible. This figure was created with BioRender.

in *E. coli* were derived from constructs containing the WT sequence by Q5 site-directed mutagenesis (NEB).

## Protein expression and affinity purification
For the purification of N-terminally His-tagged MalT, *E. coli* strain BL21 (DE3) Δ*malT220* was transformed with plasmid pOM206 or its variants, plated on LB agar containing 25 µg/ml kanamycin and grown at 37 °C overnight. A single colony was picked to prepare a starter culture

in LB containing 50 µg/ml kanamycin. 1-liter LB containing 50 µg/ml kanamycin was then inoculated with the starter culture, grown at 37 °C until OD600 ≈ 0.8 before induction by 1 mM isopropyl β-d-1-thiogalactopyranoside (IPTG) and further grown at 20 °C overnight. The bacteria were pelleted and resuspended in 100 ml TSI buffer (50 mM Tris-HCl pH 8.0, 10% sucrose, 300 mM KI) supplemented with 0.4 mM ATP or 0.1 mM ADP. Cells were disrupted by sonication on ice. After a 90-min centrifugation at 14,000 × *g*, 4 °C, the MalT protein was

purified from the supernatant by affinity purification using Ni Sepharose 6 Fast flow resin (GE Healthcare).

Expression of N-terminally GST-tagged MalY or its variants by BL21 (DE3) Δ*malT220* containing plasmid pGEX-6P-1 harboring *malY* was performed as described for MalT, except that the antibiotic was replaced by 100 μg/ml ampicillin. Cells were pelleted and resuspended in TSCl buffer (50 mM Tris-HCl pH 8.0, 10% sucrose, 300 mM KCl) supplemented with 10 μM PLP, then lysed by sonication on ice. After centrifugation, GST (Glutathione S-transferases) tagged-MalY was first purified from the supernatant by affinity purification using Glutathione Sepharose 4B resin (GE Healthcare). The GST tag is removed by enzymatic cleavage if needed.

## Gel filtration

Affinity purified proteins were subjected to gel filtration using different columns and buffers at 4 °C, the injection volume was 2 ml. MalT monomer and MalY dimer were purified using a Superdex 200 Increase 10/300 GL column (GE Healthcare) with buffer A (50 mM Tris-HCl pH 8.0, 10% sucrose, 200 mM KI, 10 mM magnesium acetate, 0.1 mM EDTA, 0.2 mM $Na_2S_2O_3$) supplemented with 0.4 mM ATP or 0.1 mM ADP and buffer TKCl (10 mM Tris-HCl pH 8.0, 100 mM KCl) supplemented with 10 μM PLP, respectively. The MalT-MalY inhibitory complex was reconstituted by incubating purified MalT (in buffer A with 0.1 mM ADP) and MalY (in buffer TKCl with 10 μM PLP) at a molar ratio of 1:1 on ice for 30 min, then purified by gel filtration using a Superdex 200 Increase 10/300 GL column (GE Healthcare) with buffer TKCl supplemented with 0.1 mM ADP and 10 μM PLP. To reconstitute MalT oligomers, purified MalT monomer was first incubated in 1×buffer I (55 mM Tris-HCl pH 8.0, 33 mM tripotassium citrate, 1 mM DTT, 1 mM maltotriose, and 0.4 mM ATP) at 20 °C for 20 minutes, then filtered through a Superose 6 Increase 10/300 GL column with buffer E (50 mM Tris-HCl pH 8.0, 50 mM NaCl, 33 mM tripotassium citrate, 1 mM DTT) supplemented with 1 mM maltotriose and 0.4 mM ATP. Monomeric WT or mutant MalT proteins for luciferase assay were first purified as described above, free nucleotides were then removed from the samples by an additional gel filtration step using buffer A without ATP and ADP.

## Pull-down assay

Affinity purified N-terminally GST-tagged MalY was incubated with an excess amount of purified WT or mutant MalT proteins on ice for 30 min in the presence of 0.1 mM ADP and 10 μM PLP, then loaded on 100 μl Glutathione Sepharose 4B resin (GE Healthcare) equilibrated with buffer TKCl supplemented with 0.1 mM ADP and 10 μM PLP. The resin was washed three times with 500 μl equilibration buffer, and proteins were eluted using the same buffer containing 20 mM glutathione (GSH). The eluted samples were analyzed by SDS-PAGE and visualized by Coomassie blue staining.

## β-galactosidase assay

A single colony from strains harboring pJM241, pJB215 or its derivatives was used to prepare a starter culture in LB containing 30 μg/ml ticarcillin. After growth at 25 °C overnight, 500 μl starter culture was used to inoculate 10 ml M9 media supplemented with 0.4% glycerol, 0.01% tryptophan, 1 μg/ml thiamine and 30 μg/ml ticarcillin, and the inoculated culture was grown at 25 °C till OD600 = 1. β-galactosidase activity was assayed as described[56]. Cells were permeabilized by chloroform and 0.02‰ SDS. Each M9 culture was assayed in triplicate and the enzymatic activity value obtained was corrected for the background as measured with pJM241 (about 11 Miller units). Fold change in β-galactosidase activity was then determined for each variant as the ratio of the variant activity to the activity of WT MalT. Means of the fold-change values were calculated from at least two independent series of cultures.

## Western blot

3 ml of M9 cell culture were centrifuged and resuspended in 160 μl 1× PBS buffer. After adding 40 μl of 5× SDS-PAGE sample buffer, samples (OD600 ≈ 15) were boiled at 100 °C for 10 min, electrophoresed using 10% SDS-PAGE gel and transferred to a nitrocellulose membrane. The membrane was blocked with TBS-T buffer containing 3% non-fat dry milk at room temperature for 2 h, probed with peroxidase-conjugated anti-HA antibody (Roche) used at 1:500 dilution overnight at 4 °C, and then revealed after incubation with 2 ml mixed SuperSignal West Dura and SuperSignal West Femto solution (Thermo Scientific).

## Luciferase assay

For nucleotide quantification, bound ADP or ATP were released from protein samples by a 10-min boiling at 92 °C and centrifugation at 14,000 × $g$ for 15 min at 4 °C. The resulting supernatant was divided equally and diluted 2.5 times in pyruvate kinase reaction buffer (125 mM Tris-HCl pH 7.4, 2.5 mM $MgSO_4$ and 5 mM phosphoenolpyruvate). 30 U pyruvate kinase (Sigma-Aldrich) (which converts ADP to ATP) or a same volume of 50% glycerol as control were added to the samples, which were then incubated at room temperature for 50 min. 100 μl of the reaction mixture was added to an equal volume of ATP Assay Mix solution (Sigma-Aldrich) and photon emissions were read immediately using a Synergy H1 microplate reader.

## EMSA

The DNA used for EMSA was a 108 bp promoter fragment containing *malPp800*, which is a mimic of the natural *pulAp* promoter and has a high binding affinity for MalT. Like *pulAp*, *malPp800* contains 4 MalT binding sites. The fragment was PCR amplified from M13malP3 *malPp800* phage[25] with Cy3 labelled primers, followed by purification using a NucleoSpin Gel and a PCR Clean-up kit (Macherey-Nagel). To reconstitute the nucleoprotein complex, promoter DNA was mixed with purified WT or mutant MalT proteins at stoichiometric ratio (1:5) in 20 μl of 1× buffer I and incubated at 20 °C for 20 min. After addition of 5× native gel sample buffer (250 mM Tris-base, 250 mM boric acid, 5 mM EDTA, 50% Glycerol, 0.1% bromophenol blue), samples were loaded on a 6% native PAGE gel (Thermo Scientific) and electrophoresed at room temperature in 0.5× TBE buffer (50 mM Tris-base, 50 mM boric acid, 1 mM EDTA). Cy3 labelled DNA was then visualized by Gel Doc XR+Imager (Bio-Rad).

## Cryo-EM sample preparation and data collection

Cryo-EM sample was prepared with purified MalT-MalY protein complex at a concentration of about 0.5 mg/ml and with MalT oligomer at 1 mg/ml. 3 μl of sample was applied to holeycarbon grids (Quantifoil Cu 1.2/1.3, 300 mesh) glow-discharged for 30 seconds at medium level in HarrickPlasma after 2 min evacuation. Grids were then blotted on filter paper (TED PELLA, INC.) for 2.5 s at 8 °C with 100% humidity and flash-frozen in liquid ethane using FEI Vitrobot Marked IV.

Titan Krios microscope operated at 300 kV and equipped with Gatan K2 Summit direct electron detector and a Gatan Quantum energy filter was used to collect micrographs. Stacks were automatically recorded using AutoEMation in super-resolution mode[57]. A nominal magnification of 64,000× was used for imaging the samples, corresponding to a final pixel size of 1.061 Å on image. Defocus values varied from −1.0 μm to −2.0 μm. Exposure rate of data collection was 23 electrons per pixel per second. The exposure time was 2.56 s dose-fractionated into 32 sub-frames, leading to a total electron exposure of ~50 electrons per Å$^2$ for each stack. An extra tilt angle of 15° was applied during MalT oligomer data collection.

## Image processing and 3D reconstruction

The flowchart of the data processing of the MalT-MalY complex is presented in Supplementary Fig. 1. All micrographs were 2 × 2 binned, generating a pixel size of 1.061 Å. Motion correction was

performed using the MotionCorr2 profram[58]. Contrast transfer function (CTF) models of dose-weighted micrographs were determined using CTFFIND4[59]. 1,854,962 particles were picked for several rounds of 2D classification using GPU-accelerated RELION[34–36]. A total of 1,145,822 good particles were selected and subjected to further 3D classification, from which 176,969 particles were selected and subject to 3D auto-refinement. CTF refinement and post-processing were performed, resulting in a final resolution of 2.94 Å for the MalT-MalY complex. The resolution was determined according to the gold-standard Fourier shell correlation 0.143 criteria with a high-resolution noise substitution method[60]. Local resolution distribution was evaluated using RELION[61].

The flowchart of the data processing of MalT oligomer was shown in Supplementary Fig. 7. 2592 micrographs were collected and preprocessed in the same way as the MalT-MalY complex dataset. In total 1,080,369 particles were extracted and subjected to 2D classification using RELION[34–36], and 834,347 good particles were used for 3D classification. Since the MalT oligomer particles had serious preferred orientation problem, the resulting density map appeared to be badly stretched.

### Model building and refinement
The crystal structure of MalY (1D2F) was docked into the EM map in Chimera[62]. Model building of MalT-MalY was manually adjusted in Coot based on the 2.94 Å reconstruction map[63]. Structural refinement was performed in PHENIX in real space with secondary structure and geometry restraints[64]. The final model after refinement was validated using Molprobity in the PHENIX package[64]. Molecular graphics and analyses were performed with ChimeraX[65,66].

### Reporting summary
Further information on research design is available in the Nature Portfolio Reporting Summary linked to this article.

## Data availability
Source data are provided with this paper. All the data needed to evaluate the conclusions of the manuscript are present in the paper or the Supplementary Information. The cryo-EM density maps and corresponding atomic coordinates for the MalT-MalY complex have been deposited in the Electron Microscopy Data Bank (EMDB) and Protein Data Bank (PDB) under accession number EMD-16140 and 8BOB. Previous PDB entries mentioned in this study include 6MFV, 1D2F, 4KRS, and 3FH6. All other data are available from the corresponding author on request. Source data are provided with this paper.

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

## Acknowledgements

Cryo-EM data were collected at the Tsinghua University Branch of the China National Center for Protein Sciences (Beijing). We acknowledge support with computational facilities from the Bio-Computing Plat-form cluster. We thank Xiaoxiao Zhang (ShanghaiTech University) for assisting cryo-EM data acquisition and processing. This work was supported by the Alexander von Humboldt Foundation (a Humboldt professorship to J.C.), the Max-Planck-Gesellschaft (a Max Planck fellowship to J.C.), and the National Natural Science Foundation of China (31421001 to J.C.).

## Author contributions

J.C., Y.W., and E.R. designed the project. J.C. supervised the project. Y.W. performed the biochemical and functional experiments, and prepared the figures. Y.S. processed cryo-EM data and produced the final map. Y.S. and J.C. built the complex model. E.R. collected materials for strain construction. J.C. and Y.W. wrote the manuscript with input from E.R., Z.H., and Y.S.

## Funding

## Competing interests

The authors declare no competing interests.
