## [Peer Review File · Nature Communications]

Structural basis for negative regulation of the Escherichia coli maltose systemREVIEWER COMMENTS

Reviewer #1 (Remarks to the Author):

Proteins in the signal transduction ATPase with numerous domains (STAND) family are involved in the wide variety of cellular process from prokaryote to eukaryote. In this study, the authors focus on the bacterial STAND protein MalT, which regulate E. coli maltose metabolism and is negatively regulated by MalY. The cryo-EM structure of MalT in complex with MalY reveals that MalY and MalT forms a 2:2 complex through MalY binding to the putative self-oligomerization surface of MalT and thereby prevents self-oligomerization of MalT for downstream activation. The authors also claim that the N-terminal region of MalT exhibits dual regulatory roles for both autoinhibition and activation.

Overall, the inhibitory mechanism of MalT by MalY presented here will contribute to our understanding of how STAND family proteins are regulated. While the data presented generally supports the authors' conclusion, some points require clarification before I can fully recommend its publication.

Major points:

1. Roles of N-loop

The authors claim, based on the data presented in Figures 4 and 5, that the N-loop of MalT exhibits dual functions in both autoinhibition and activation. However, some aspects of the data presented are unclear. Figure 4B depicts that all of the mutant proteins exhibit a reduction in oligomerization compared to the wild-type protein, but some of the mutations were found to enhance transcriptional activity in Figure 3B, implying that they ought to result in enhanced self-oligomerization. Could the authors clarify this apparent inconsistency?

The authors analyzed the amount of nucleotides bound to both wild type (WT) and mutant proteins (Fig. 3C and Fig. 4C). In Fig. 3C, they stated that the increased ADP/ATP exchange led to increased activity for the S5L and R9S mutations. However, K9A and R19A mutations also showed increased ADP/ATP exchange in Fig. 4C but in this case these mutations led to the decreased activity. It appears that there is no clear correlation between the ATP levels in these assays and MalT activity. It is also not clear whether the nucleotide content in these data reflects reduced ATP hydrolysis or simply increased ADP dissociation.

2. Cryo-EM analysis of MalT oligomer

The authors successfully reconstituted active MalT oligomer and showed clear cryo-EM 2D images of MalT oligomer in Fig. 6A. This is very important data supporting the conclusion of this paper but the author did not provide any details for this data. How was the oligomer sample prepared and how were the cryo-EM data acquired and analyzed? The original cryo-EM micrograph should be shown. The authors showed cryo-EM 2D images of MalT-MalY complex for reference (Fig. 6B), but this is not an appropriate control because the presence of MalY makes comparison difficult. Cryo-EM of MalT monomer would be an appropriate control.

The authors stated that oligomer of MalT showed preferred orientations problem and failed to reconstruct 3D map. It is certainly difficult to reconstruct a high-quality 3D map from data with limited angular distribution, but in many cases, it is still possible to reconstruct a 3D map, although it would be of low quality. Nonetheless, if possible, it would be useful to check if the oligomer model of open NOD modules (e.g. Fig. 5D) fits this map. It would also be beneficial, though not essential, to collect tilt data.

Minor points:

1. The authors describes the similarity between MalT and its homolog PH0952 from *Pyrococcus horikoshii* (line 103), but it is difficult to see the similarity by comparing MalT (Fig. 1 or Fig. 2) and PH0952 (Extended Data Fig. 2) in different figures with different orientations. The author should provide a side-by-side comparison or superposition.

2. In Fig. 1B and Fig. 2A, each protomer needs to be labeled.

3. In line 161 "By contrast ... (data not shown)", data should be shown.

4. In Fig. 3D, the order of SDS-PAGE (right) should be arranged similar to the legend of the chart

on the left.

5. In line 186, "the WHD and the NBD" would be "the WHD, HD1, and NBD".

6. In line 186, the authors stated that the N-loop segment of inactive PH0952 forms much fewer contacts with the two structural domains (Extended Data Fig. 2). However, it seems that the N-loop of PH0952 interacts with HD1, WHD, and NBD (Extended Data Fig. 2). A figure showing a direct comparison between the two structures would be needed to discuss it.

7. In line 189, the authors stated that there is no interaction between ADP and WHD in MalT. The conserved histidine (WHD)-mediated interaction with phosphate group of ADP plays a role for maintaining inactive conformation of NOD in many NOD proteins structurally characterized to date. The authors need to clarify whether the lack of the interaction between WHD and ADP is due to an altered arrangement of the NOD domain or simply due to the absence of the corresponding His residue in MalT.

8. Line 195 "Though these two residues ... the N-loop" does not make sense. The size of the side chains of M311I and W317P (Ref. 31) and W317R (Ref. 28) are similar to or smaller than the original.

9. In line 225, data should be shown. This is an important result.

10. In line 316 to 319 "Consistent with this model, ... (Extended Data Fig. 6b).", this should be moved to the result section.

Reviewer #2 (Remarks to the Author):

The manuscript by Wu et al. describes the cryo-EM structure of the transcriptional activator MalT in complex with the PLP dependent enzyme MalY, which inhibits MalT transcriptional activity. Bacterial MalT is one of the best studied examples of a signal transduction ATPase with numerous domains (STAND) family member, to which important signaling proteins involved in eukaryotic programmed cell death and innate immunity proteins (e.g. Apaf-1, CED-4, ZAR1 etc.) belong to. Similar to other STAND proteins, MalT autoinhibition and transcriptional activation is orchestrated by the ADP / ATP content and ATP hydrolysis. Additionally, maltotriose activates MalT and three protein factors (MalK, Aes, and MalY) antagonize transcriptional activation. No structural information for any of those complexes has been reported and a detailed understanding of their mode of MalT inhibition was elusive so far. Given the importance of STAND proteins and the pivotal role of MalT within this family, the reported findings – if further validated – could be of great interest to the field.

The authors provide a high-resolution cryo-EM structure revealing that dimeric MalY is sandwiched between two MalT protomers present in autoinhibited ADP bound conformation. Furthermore, they provide evidence that MalY shields an important surface patch on MalT, thereby inhibiting self-oligomerization and as consequence, transcriptional activity. Interestingly, the corresponding interaction patch on MalY had been mapped by previous mutational studies, and the reported structure nicely corroborates this interface. Furthermore, the authors show that a loop region N-terminal to the nucleotide binding and oligomerization domain (NOD) is sandwiched between the latter on one side and the helical domain (HD), the winged-helix-domain (WHD) arm domain juxtaposed to it. This region has already been implemented in regulating autoinhibition and the authors provide here additional biochemical and in vivo evidence supporting its potential role in regulating MalT.

In general, most experiments are convincing, and the conclusions drawn by the authors are justified. However, the manuscript contains ambiguities and caveats in key findings that require further experimental verification.

Major Points

1) The authors claim to determine nucleotide exchange rates of wild type MalT and selected mutants. To this end, they incubated MalT in a functional ATPase assay (using ATP.Mg²⁺ and maltotriose) for 20 min, performed size exclusion in a nucleotide-free buffer and monitored the bound nucleotide by determining ATP/ADP levels liberated from the protein. However, the assumption made by the authors is only correct if all variants have the same enzymatic properties and thus have comparable ATP/ADP ratios thorough the course of the experiment. There is in fact strong evidence from literature that self-assembled MalT presents mainly the ATP-hydrolysis competent oligomeric form. Thus, the authors cannot exclude that mutated variants have different enzymatic properties as these may substantially differ in their self-oligomerization properties. Is this the case?

In sum, the conclusions drawn upon analyzing ATP and ADP levels appear to be preliminary and do not provide compelling evidence in order to compare nucleotide content (including ATP/ADP ratios) and the oligomeric state. For instance, the authors suggest that increased levels of ATP against ADP for variant K5A (Figure 5C) are due to its monomeric and thus, potentially autoinhibited state. In contrast, the S5L variant (Figure 4C) has similar ATP/ADP ratio as K5A, is not monomeric and seems to have similar self-oligomerization properties as the R12L mutant. For the latter, however, the ATP/ADP levels are significantly different than for S5L. Furthermore, the authors state in the manuscript that the R19A variant has reduced nucleotide affinity (and depending on the experimental variance, see comment below, most likely this is also the case for K6A) which suggest that indeed catalysis is impaired. Could it be, that these ambiguities complicating any comparison of nucleotide content with oligomerization properties and transcription activation are due to different catalytic properties and not ATP/ADP exchange? Wouldn't it be better to uncouple ATP hydrolysis from ATP/ADP exchange in their experiments by using non-hydrolysable ATP analogs? Or at least make sure that ATP/ADP concentrations are above saturation and identical in all experiments before size exclusion. Addressing these separate points would be most critical to provide a conclusive model.

2) No experimental errors are given for nucleotide content analysis. Are these single measurements? Clearly, such experiments need to be performed in experimental triplicates. Furthermore, since wild type and many protein variants seem to elute as multimeric species, the choice of fraction(s) used for analyzing the nucleotide content is important. Can the authors exclude that different nucleotide content or ratios are simply artifacts arising from dilution and nucleotide dissociation during size exclusion chromatography?

Minor Points

1) A rather small surface patch of MalT interacts with MalY. Mutation of R173 to A in MalT abolishes interaction with MalY, uncouples transcriptional activity from MalY and causes reduce self-oligomerization of MalT, verifying that the observed assembly is correct. A residue in proximity, R171, has been suggested to be involved in MalT self-oligomerization and ATP hydrolysis (Marquet & Richet, 2010 doi 10.1128/JB.00522-10). Given that R171 is located at the C-terminus of alpha8, it would be worth showing the interaction network of R171 in Figure 2A and discuss its potential role in the manuscript.

2) The interaction network of MalY with MalT has previously been mapped by MalT truncation experiments (Schlegel et al., 2002, doi 10.1128/JB.184.11.3069-3077.2002). Interaction of a MalT fragment encoding the N-terminal loop, NBD, and a fragment of the HD domain with MalY was shown to be much weaker when compared with one additionally containing the WHD domain. Yet, the Cryo-EM structure reveals that just the NBD interacts with MalY. Do the authors have an explanation for that observation?

3) It is not clear which nomenclature is used when referring to alpha-helix 7 and 8 in the manuscript. The authors might want to consider showing an illustration of MalT-topology or mention which nomenclature for secondary structure elements in STAND proteins they are referring to.

4) In Figure 5A, R19 should read as R19A

5) Starting with line 80, the authors question the relevance of a previously proposed interaction of the NBD and the sensor domain, focusing exclusively on the lack of a defined cryoEM-map feature. This negative argumentation is not fully justified and needs further experimental proof/hints. Furthermore, with regards to the presented data, what is the local resolution of the NBD in that region in comparison to the rest of the NBD?

Reviewer #1 (Remarks to the Author):

Proteins in the signal transduction ATPase with numerous domains (STAND) family are involved in the wide variety of cellular process from prokaryote to eukaryote. In this study, the authors focus on the bacterial STAND protein MalT, which regulate E. coli maltose metabolism and is negatively regulated by MalY. The cryo-EM structure of MalT in complex with MalY reveals that MalY and MalY forms a 2:2 complex through MalY binding to the putative self-oligomerization surface of MalT and thereby prevents self-oligomerization of MalT for downstream activation. The authors also claim that the N-terminal region of MalT exhibits dual regulatory roles for both autoinhibition and activation.

Overall, the inhibitory mechanism of MalT by MalY presented here will contribute to our understanding of how STAND family proteins are regulated. While the data presented generally supports the authors' conclusion, some points require clarification before I can fully recommend its publication.

Major points:

1. Roles of N-loop

The authors claim, based on the data presented in Figures 4 and 5, that the N-loop of MalT exhibits dual functions in both autoinhibition and activation. However, some aspects of the data presented are unclear. Figure 4B depicts that all of the mutant proteins exhibit a reduction in oligomerization compared to the wild-type protein, but some of the mutations were found to enhance transcriptional activity in Figure 3B, implying that they ought to result in enhanced self-oligomerization. Could the authors clarify this apparent inconsistency?

Thank you for pointing this out. The manuscript has been modified accordingly to clarify the issue.

The inconsistency between the enhanced *in vivo* transcriptional activities of the variants and their reduced abilities to oligomerize *in vitro* is only apparent. The pathway leading to transcription activation by MalT is a complex process that comprises inducer binding, NOD opening, nucleotide exchange, protein self-association, promoter binding, and RNA polymerase recruitment. The rate-limited step might differ depending on the step of the process being monitored, the assay conditions, and the step altered by the mutation. In particular, the *in vitro* oligomerization assays are performed in the presence of saturating concentration of ATP and inducer, but in the absence of ADP. Under these conditions that favor NOD opening and nucleotide exchange, the assay is insensitive to differences in stability of the closed-NOD state, and reflects the intrinsic ability of the variant to self-associate. In contrast, when transcriptional activities are monitored *in vivo*, i.e., in the presence of both ADP and ATP as well as a low level of endogenous maltotriose, NOD opening is the rate-limited step as exemplified by the variant MalT-W317R, of which the NOD is locked in an open state and is 33-fold more active than WT (Liu et al., Mol Microbiol, 2013, doi: 10.1111/mmi.12434). Consequently, the enhanced activities of the S5L and R9S variants result from the facilitation of NOD opening by the substitutions, with the benefice of autoinhibition relief more than compensating the oligomerization defect. This interpretation is further supported by the observation that the S5L, S8A, and R9S variants still bind promoter DNA despite a smaller size of oligomers formed

compared to WT (Extended Data Fig. 8), with cooperative binding of the MalT-binding sites present in the promoter stabilizing the oligomers.

The authors analyzed the amount of nucleotides bound to both wild type (WT) and mutant proteins (Fig. 3C and Fig. 4C). In Fig. 3C, they stated that the increased ADP/ATP exchange led to increased activity for the S5L and R9S mutations. However, K9A and R19A mutations also showed increased ADP/ATP exchange in Fig. 4C but in this case these mutations led to the decreased activity. It appears that there is no clear correlation between the ATP levels in these assays and MalT activity. It is also not clear whether the nucleotide content in these data reflects reduced ATP hydrolysis or simply increased ADP dissociation.

Thanks for pointing this issue out. We believe that this question might result from our unclear description of the text.

An increased ADP/ATP exchange indicates a destabilized NOD module and lessened autoinhibition mediated by the N-loop. This is expected to lead to a higher transcriptional activity if the MalT variant still retains the activities of oligomerization and DNA binding, as shown for MalT R9S. However, both MalT K6A and R19A are no longer able to self-associate, which explains abolition of their transcriptional activity.

Hydrolysis of ATP by MalT is dependent on MalT oligomerization. Maltotriose, which is required for MalT oligomerization, was absent in the buffers used for protein purification and for the last step of gel filtration that removes free ATP before nucleotide quantification. In addition, the level of ATP hydrolysis uncoupled from MalT oligomerization is lower than 10% of that measured in the presence of maltotriose (Marquenet & Richet, *J Bacteriol*, 2010, doi: 10.1128/JB.00522-10). Furthermore, all of these chromatography steps were performed at 4 °C and in the presence of 200 mM potassium iodide, conditions in which the ATPase activity of MalT should be limited. Thus, the ATP hydrolysis activity of monomeric MalT proteins used for the nucleotide quantification assays is expected to be very low. As a result, the enhanced ATP/ADP ratio observed for some variants should result from destabilization of the closed-NOD state rather than a reduced ATPase activity. To avoid ambiguity, the original text describing the sample preparation steps for the luciferase assay has been modified in the revised manuscript.

2. Cryo-EM analysis of MalT oligomer

The authors successfully reconstituted active MalT oligomer and showed clear cryo-EM 2D images of MalT oligomer in Fig. 6A. This is very important data supporting the conclusion of this paper but the author did not provide any details for this data. How was the oligomer sample prepared and how were the cryo-EM data acquired and analyzed? The original cryo-EM micrograph should be shown. The authors showed cryo-EM 2D images of MalT-MalY complex for reference (Fig. 6B), but this is not an appropriate control because the presence of MalY makes comparison difficult. Cryo-EM of MalT monomer would be an appropriate control.

The authors stated that oligomer of MalT showed preferred orientations problem and failed to reconstruct 3D map. It is certainly difficult to reconstruct a high-quality 3D map from data with limited angular distribution, but in many cases, it is still possible to reconstruct a 3D map, although it would be of low quality. Nonetheless, if possible, it would be useful to check if the oligomer model of open NOD modules (e.g. Fig. 5D) fits this map. It would also be beneficial, though not essential, to collect tilt data.

Thanks for the constructive suggestion. Descriptions of protein preparation and cryo-EM analysis of oligomeric samples have been added in Methods section and Extended Data Fig. 7. We did collect tilt data, but unfortunately it was not very helpful for 3D reconstruction of the oligomeric samples. We also tried using different types of grids (e. g. copper grid coated with lacy carbon film) and adding cross-linking agents during sample preparation (e. g. glutaraldehyde), but the preferred orientation problem was still not resolved. Nevertheless, a preliminary 3D map was reconstructed from the tilt data collected. The structure of monomeric MalT in its active conformation (without DNA-binding domain) was predicted using Alphafold2, and the modeled structure fitted well into the protomer density from the 3D map, suggesting the extra density observed for oligomeric MalT samples is likely from the C-terminal domains of MalT. We do not have the cryo-EM data from the MalT monomer sample. The cryo-EM 2D images of MalT-MalY complex are removed from Fig. 6 as suggested.

Minor points:

1. The authors describes the similarity between MalT and its homolog PH0952 from *Pyrococcus horikoshii* (line 103), but it is difficult to see the similarity by comparing MalT (Fig. 1 or Fig. 2) and PH0952 (Extended Data Fig. 2) in different figures with different orientations. The author should provide a side-by-side comparison or superposition.

Change has been made in Extended Data Fig. 3 as suggested.

2. In Fig. 1B and Fig. 2A, each protomer needs to be labeled.

Change has been made as suggested.

3. In line 161 “By contrast ... (data not shown)”, data should be shown.

This data is now shown in Extended Data Fig. 5.

4. In Fig. 3D, the order of SDS-PAGE (right) should be arranged similar to the legend of the chart on the left.

Change has been made as suggested.

5. In line 186, “the WHD and the NBD” would be “the WHD, HD1, and NBD”.

Change has been made as suggested.

6. In line 186, the authors stated that the N-loop segment of inactive PH0952 forms much fewer contacts with the two structural domains (Extended Data Fig. 2). However, it seems that the N-loop of PH0952 interacts with HD1, WHD, and NBD (Extended Data Fig. 2). A figure showing a direct comparison between the two structures would be needed to discuss it.

Change has been made in Extended Data Fig. 3 as suggested.

7. In line 189, the authors stated that there is no interaction between ADP and WHD in MalT. The conserved histidine (WHD)-mediated interaction with phosphate group of ADP plays a role for maintaining inactive conformation of NOD in many NOD proteins structurally characterized to date. The authors need to clarify whether the lack of the interaction between WHD and ADP is due to an altered arrangement of the NOD domain or simply due to the absence of the corresponding His residue in MalT.

We thank the reviewer for pointing this out. As commented by the reviewer, ADP is shown important for the inactive conformation of many NOD proteins through hydrogen bonding with the conserved histidine residue from the WHD. Structural alignment of MalT with other NLRs (e. g. NLRC4, Apaf-1) showed that this conserved histidine also exists in MalT (H321). In the finally refined structure, the bound ADP molecule indeed hydrogen bonds with conserved H321 via its phosphate group, and the distance between H321 and the bound ADP is about 3.4 Å. The text describing the changes has been added to the revised manuscript accordingly.

8. Line 195 “Though these two residues ... the N-loop” does not make sense. The size of the side chains of M311I and W317P (Ref. 31) and W317R (Ref. 28) are similar to or smaller than the original.

Change has been made as suggested.

9. In line 225, data should be shown. This is an important result.

The data has been added in the revision.

Deletion of the first 8 amino acids of MalT (hereafter called Ndel8) results in loss of transcriptional activity. However, as shown in (a), the variant protein cannot be detected by western blot probably due to its lowered stability under cellular conditions. Nevertheless, like variant K6A, Ndel8 protein can be purified *in vitro* and a gel filtration analysis shows that it remains monomeric in the presence of maltotriose (b). A nucleotide quantification assay shows that the Ndel8 is characterized by a high ATP/ADP ratio (c), as observed with variants K6A and R19A. Therefore, we believe that the loss of function phenotype of Ndel8 is mainly due to its inability to form oligomers for DNA-binding and that a monomer with an open NOD is unstable *in vivo*. We have added this piece of data and modified the text as suggested.

10. In line 316 to 319 “Consistent with this model, ... (Extended Data Fig. 6b).”, this should be moved to the result section.

Thanks for the suggestion. Change has been made as suggested.

Reviewer #2 (Remarks to the Author):

The manuscript by Wu et al. describes the cryo-EM structure of the transcriptional activator MalT in complex with the PLP dependent enzyme MalY, which inhibits MalT transcriptional activity. Bacterial MalT is one of the best studied examples of a signal transduction ATPase with numerous domains (STAND) family member, to which important signaling proteins involved in eukaryotic programmed cell death and innate immunity proteins (e.g. Apaf-1, CED-4, ZAR1 etc.) belong to. Similar to other STAND proteins, MalT autoinhibition and transcriptional activation is orchestrated by the ADP / ATP content and ATP hydrolysis. Additionally, maltotriose activates MalT and three protein factors (MalK, Aes, and MalY) antagonize transcriptional activation. No structural information for any of those complexes has been reported and a detailed understanding of their mode of MalT inhibition was elusive so far. Given the importance of STAND proteins and the pivotal role of MalT within this family, the reported findings – if further validated – could be of great interest to the field.

The authors provide a high-resolution cryo-EM structure revealing that dimeric MalY is sandwiched between two MalT protomers present in autoinhibited ADP bound conformation. Furthermore, they provide evidence that MalY shields an important surface patch on MalT, thereby inhibiting self-oligomerization and as consequence, transcriptional activity. Interestingly, the corresponding interaction patch on MalY had been mapped by previous mutational studies, and the reported structure nicely corroborates this interface. Furthermore, the authors show that a loop region N-terminal to the nucleotide binding and oligomerization domain (NOD) is sandwiched between the latter on one side and the helical domain (HD), the winged-helix-domain (WHD) arm domain juxtaposed to it. This region has already been implemented in regulating autoinhibition and the authors provide here additional biochemical and in vivo evidence supporting its potential role in regulating MalT.

In general, most experiments are convincing, and the conclusions drawn by the authors are justified. However, the manuscript contains ambiguities and caveats in key findings that require further experimental verification.

Major Points

1) The authors claim to determine nucleotide exchange rates of wild type MalT and selected mutants. To this end, they incubated MalT in a functional ATPase assay (using ATP.Mg²⁺ and maltotriose) for 20 min, performed size exclusion in a nucleotide-free buffer and monitored the bound nucleotide by determining ATP/ADP levels liberated from the protein. However, the assumption made by the authors is only correct if all variants have the same enzymatic properties and thus have comparable ATP/ADP ratios thorough the course of the experiment.

There is in fact strong evidence from literature that self-assembled MalT presents mainly the ATP-hydrolysis competent oligomeric form. Thus, the authors cannot exclude that mutated variants have different enzymatic properties as these may substantially differ in their self-oligomerization properties. Is this the case?

Reviewer 1 raised a similar question, which we believe arise from our unclear description of the preparation of the protein samples used for the nucleotide quantification assay. The MalT variants were purified following the standard procedure as described in Methods section, with the last step corresponding to a gel filtration in the presence of 0.4 mM ATP only. All the variants eluted as monomers under these conditions. Free ATP was then removed by gel filtration before quantification of the MalT-bound nucleotides, as described previously (Marquenet & Richet, Mol Cell, 2007, doi: 10.1016/j.molcel.2007.08.014). Maltotriose, which is required for MalT oligomerization, was not used during the sample preparation steps. Thus, the nucleotide quantification assay was performed on monomeric MalT proteins, of which the ATPase activity is expected to be very low (see the answer to major point 1 from Reviewer 1). To avoid ambiguity, the original text describing the sample preparation steps for luciferase assay has been modified.

In sum, the conclusions drawn upon analyzing ATP and ADP levels appear to be preliminary and do not provide compelling evidence in order to compare nucleotide content (including ATP/ADP ratios) and the oligomeric state. For instance, the authors suggest that increased levels of ATP against ADP for variant K5A (Figure 5C) are due to its monomeric and thus, potentially autoinhibited state. In contrast, the S5L variant (Figure 4C) has similar ATP/ADP ratio as K5A, is not monomeric and seems to have similar self-oligomerization properties as the R12L mutant. For the latter, however, the ATP/ADP levels are significantly different than for S5L. Furthermore, the authors state in the manuscript that the R19A variant has reduced nucleotide affinity (and depending on the experimental variance, see comment below, most likely this is also the case for K6A) which suggest that indeed catalysis is impaired. Could it be, that these ambiguities complicating any comparison of nucleotide content with oligomerization properties and transcription activation are due to different catalytic properties and not ATP/ADP exchange? Wouldn't it be better to uncouple ATP hydrolysis from ATP/ADP exchange in their experiments by using non-hydrolysable ATP analogs? Or at least make sure that ATP/ADP concentrations are above saturation and identical in all experiments before size exclusion. Addressing these separate points would be most critical to provide a conclusive model.

We thank the reviewer for his/her comments. As mentioned in our discussion addressing the previous question, the nucleotide quantification assays were performed in the absence of maltotriose required for MalT ATPase activity.

The N-terminal fragment has a dual role in mediating MalT activation by participating in autoinhibition and oligomerization. We used nucleotide quantification and gel filtration assays to investigate impact of mutations in this region on autoinhibition and oligomerization of MalT, respectively. The proteins were purified using gel filtration in the presence of a saturating concentration of ATP (0.4 mM), and peak fractions corresponding to the monomeric position of MalT were pooled for both assays. The nucleotide quantification assays were performed as described above. As shown in Figs. 4c and 5c, mutations of S5, R9, K6, R19 resulted in ATP binding by MalT, indicating they have a role in MalT autoinhibition. To test the effect of the N-terminal fragment mutations on MalT oligomerization, the pooled proteins were characterized using gel filtration in the presence of ATP and maltotriose. The K6A and R19A mutations completely disrupted MalT oligomerization induced by maltotriose, which can explain the loss-of-function of these two mutants (Fig. 5c). Compared to K6A, MalT S5L still retained activity of maltotriose-induced oligomerization and consequent transcriptional activity. R12A

had no effect on MalT autoinhibition but compromised its oligomerization to a similar extent as S5L, and has no impact on the *in vivo* activity of this mutant.

2) No experimental errors are given for nucleotide content analysis. Are these single measurements? Clearly, such experiments need to be performed in experimental triplicates. Furthermore, since wild type and many protein variants seem to elute as multimeric species, the choice of fraction(s) used for analyzing the nucleotide content is important. Can the authors exclude that different nucleotide content or ratios are simply artifacts arising from dilution and nucleotide dissociation during size exclusion chromatography?

Thanks for pointing this out. As suggested by the reviewer, the nucleotide content of different protein samples was indeed measured in triplicates. Fig. 4c has been modified and the information of experimental errors has been added in the revision.

As explained above, the nucleotide content was measured on monomeric proteins.

Nucleotide quantification was performed using the established method (Marquenet & Richet, Mol Cell, 2007, doi: 10.1016/j.molcel.2007.08.014). The mutagenesis data of nucleotide content was consistent with the cryo-EM structure. Ligand binding is required for NLRs to exchange of ADP with ATP, which is also true with MalT. This indicates that ADP dissociation from at least wild type MalT is less likely to happen in our gel filtration conditions containing no maltotriose. Mutant proteins may have a lower nucleotide-binding affinity and the possibility of nucleotide dissociation cannot be excluded (e. g. S5L, K6A, and R19A). However, if this does happen, the data actually confirm inhibitory effect of these mutated positions on nucleotide exchange of MalT.

Minor Points

1) A rather small surface patch of MalT interacts with MalY. Mutation of R173 to A in MalT abolishes interaction with MalY, uncouples transcriptional activity from MalY and causes reduce self-oligomerization of MalT, verifying that the observed assembly is correct. A residue in proximity, R171, has been suggested to be involved in MalT self-oligomerization and ATP hydrolysis (Marquenet & Richet, 2010 doi 10.1128/JB.00522-10). Given that R171 is located at the C-terminus of alpha8, it would be worth showing the interaction network of R171 in Figure 2A and discuss its potential role in the manuscript.

Change has been made as suggested.

2) The interaction network of MalY with MalT has previously been mapped by MalT truncation experiments (Schlegel et al., 2002, doi 10.1128/JB.184.11.3069-3077.2002). Interaction of a MalT fragment encoding the N-terminal loop, NBD, and a fragment of the HD domain with MalY was shown to be much weaker when compared with one additionally containing the WHD domain. Yet, the Cryo-EM structure reveals that just the NBD interacts with MalY. Do the authors have an explanation for that observation?

The WHD of MalT may facilitate MalT-MalY interaction indirectly by stabilizing the conformation of the fragment encompassing NBD and part of HD. Additionally, it remains undetermined how deletion of the WHD impacts folding of the MalT fragment. It could be that it was only partially folded when expressed *in vivo*.

3) It is not clear which nomenclature is used when referring to alpha-helix 7 and 8 in the manuscript. The authors might want to consider showing an illustration of MalT-topology or mention which nomenclature for secondary structure elements in STAND proteins they are referring to.

Numbering of alpha helices was based on their order (from the N- to the C-terminal end) in the structure. A structure with secondary structural element labeled has been added to the revision (Extended Data Fig. 2).

4) In Figure 5A, R19 should read as R19A

Change has been made as suggested.

5) Starting with line 80, the authors question the relevance of a previously proposed interaction of the NBD and the sensor domain, focusing exclusively on the lack of a defined cryoEM-map feature. This negative argumentation is not fully justified and needs further experimental proof/hints. Furthermore, with regards to the presented data, what is the local resolution of the NBD in that region in comparison to the rest of the NBD?

Based on our structural findings, the interaction between NBD and the sensor domain of MalT seems to be not essential for MalY inhibition/autoinhibition, but we did not exclude the possibility that such a contact exists during other steps of MalT regulation, as mentioned in the last section of our discussion. The local resolution of the NBD surface (near residues M96, Y102) that presumably interacts with the sensor is similar to other parts of NBD, which is between 3-3.5 Å.

REVIEWERS' COMMENTS

Reviewer #1 (Remarks to the Author):

The authors have satisfactorily addressed most of the concerns or suggestions I raised in the initial manuscript and the revised manuscript is substantially improved. I have no further comments.

Reviewer #2 (Remarks to the Author):

In this revised version and the rebuttal letter, the authors have carefully addressed all concerns raised by the reviewers and I would like to congratulate the authors to their achievements.

I do only have one minor comment, which the authors might consider in the final manuscript: I guess that the authors assume that purified MalT is stoichiometrically bound to ADP after purification similar as reported by Marquet and Richet, in 2007. It would be worth explicitly mentioning this in the manuscript when describing their experiments to determine ADP/ATP exchange rates, in order to avoid any misunderstanding.

REVIEWERS' COMMENTS

Reviewer #1 (Remarks to the Author):

The authors have satisfactorily addressed most of the concerns or suggestions I raised in the initial manuscript and the revised manuscript is substantially improved. I have no further comments.

We again thank the reviewer for the suggestions, based on which we were able to make improvements in the manuscript section discussing MalT activation mechanism.

Reviewer #2 (Remarks to the Author):

In this revised version and the rebuttal letter, the authors have carefully addressed all concerns raised by the reviewers and I would like to congratulate the authors to their achievements.

I do only have one minor comment, which the authors might consider in the final manuscript: I guess that the authors assume that purified MalT is stoichiometrically bound to ADP after purification similar as reported by Marquet and Richet, in 2007. It would be worth explicitly mentioning this in the manuscript when describing their experiments to determine ADP/ATP exchange rates, in order to avoid any misunderstanding.

We thank the review for the comment. The paper by Marquet and Richet is now cited in the section describing the nucleotide exchange assay, which makes the context clearer.